# Multi-output Polynomial Networks and Factorization Machines

**Mathieu Blondel**
NTT Communication Science Laboratories
Kyoto, Japan
mathieu@mblondel.org

**Vlad Niculae**[*]
Cornell University
Ithaca, NY
vlad@cs.cornell.edu

**Takuma Otsuka**
NTT Communication Science Laboratories
Kyoto, Japan
otsuka.takuma@lab.ntt.co.jp

**Naonori Ueda**
NTT Communication Science Laboratories
RIKEN
Kyoto, Japan
ueda.naonori@lab.ntt.co.jp

## Abstract

Factorization machines and polynomial networks are supervised polynomial models based on an efficient low-rank decomposition. We extend these models to the multi-output setting, i.e., for learning vector-valued functions, with application to multi-class or multi-task problems. We cast this as the problem of learning a 3-way tensor whose slices share a common basis and propose a convex formulation of that problem. We then develop an efficient conditional gradient algorithm and prove its global convergence, despite the fact that it involves a non-convex basis selection step. On classification tasks, we show that our algorithm achieves excellent accuracy with much sparser models than existing methods. On recommendation system tasks, we show how to combine our algorithm with a reduction from ordinal regression to multi-output classification and show that the resulting algorithm outperforms simple baselines in terms of ranking accuracy.

## 1 Introduction

Interactions between features play an important role in many classification and regression tasks. Classically, such interactions have been leveraged either explicitly, by mapping features to their products (as in polynomial regression), or implicitly, through the use of the kernel trick. While fast linear model solvers have been engineered for the explicit approach [9, 28], they are typically limited to small numbers of features or low-order feature interactions, due to the fact that the number of parameters that they need to learn scales as $O(d^t)$, where $d$ is the number of features and $t$ is the order of interactions considered. Models kernelized with the polynomial kernel do not suffer from this problem; however, the cost of storing and evaluating these models grows linearly with the number of training instances, a problem sometimes referred to as the curse of kernelization [30].

Factorization machines (FMs) [25] are a more recent approach that can use pairwise feature interactions efficiently even in very high-dimensional data. The key idea of FMs is to model the weights of feature interactions using a **low-rank** matrix. Not only this idea offers clear benefits in terms of model compression compared to the aforementioned approaches, it has also proved instrumental in modeling interactions between **categorical** variables, converted to binary features via a one-hot encoding. Such binary features are usually so sparse that many interactions are never observed in the

---

[*]Work performed during an internship at NTT Commmunication Science Laboratories, Kyoto.

training set, preventing classical approaches from capturing their relative importance. By imposing a low rank on the feature interaction weight matrix, FMs encourage shared parameters between interactions, allowing to estimate their weights even if they never occurred in the training set. This property has been used in recommender systems to model interactions between user variables and item variables, and is the basis of several industrial successes of FMs [32, 17].

Originally motivated as neural networks with a polynomial activation (instead of the classical sigmoidal or rectifier activations), polynomial networks (PNs) [20] have been shown to be intimately related to FMs and to only subtly differ in the non-linearity they use [5]. PNs achieve better performance than rectifier networks on pedestrian detection [20] and on dependency parsing [10], and outperform kernel approximations such as the Nyström method [5]. However, existing PN and FM works have been limited to single-output models, i.e., they are designed to learn scalar-valued functions, which restricts them to regression or binary classification problems.

**Our contributions.** In this paper, we generalize FMs and PNs to multi-output models, i.e., for learning vector-valued functions, with application to multi-class or multi-task problems.

1) We cast learning multi-output FMs and PNs as learning a 3-way tensor, whose slices **share a common basis** (each slice corresponds to one output). To obtain a **convex formulation** of that problem, we propose to cast it as learning an infinite-dimensional but **row-wise sparse** matrix. This can be achieved by using group-sparsity inducing penalties. (§3)

2) To solve the obtained optimization problem, we develop a variant of the **conditional gradient** (a.k.a. Frank-Wolfe) algorithm [11, 15], which repeats the following two steps: i) select a new basis vector to add to the model and ii) refit the model over the current basis vectors. (§4) We prove the **global convergence** of this algorithm (Theorem 1), despite the fact that the basis selection step is non-convex and more challenging in the shared basis setting. (§5)

3) On multi-class classification tasks, we show that our algorithm achieves comparable accuracy to kernel SVMs but with much more compressed models than the Nyström method. On recommender system tasks, where kernelized models cannot be used (since they do not generalize to unseen user-item pairs), we demonstrate how our algorithm can be combined with a reduction from ordinal regression to multi-output classification and show that the resulting algorithm **outperforms single-output PNs and FMs** both in terms of root mean squared error (RMSE) and **ranking accuracy**, as measured by nDCG (normalized discounted cumulative gain) scores. (§6)

## 2 Background and related work

**Notation.** We denote the set $\{1, \ldots, m\}$ by $[m]$. Given a vector $\boldsymbol{v} \in \mathbb{R}^k$, we denote its elements by $v_r \in \mathbb{R} \ \forall r \in [k]$. Given a matrix $\boldsymbol{V} \in \mathbb{R}^{k \times m}$, we denote its rows by $\boldsymbol{v}_r \in \mathbb{R}^m \ \forall r \in [k]$ and its columns by $\boldsymbol{v}_{:,c} \ \forall c \in [m]$. We denote the $l_p$ norm of $\boldsymbol{V}$ by $\|\boldsymbol{V}\|_p := \|\operatorname{vec}(\boldsymbol{V})\|_p$ and its $l_p/l_q$ norm by $\|\boldsymbol{V}\|_{p,q} := \left( \sum_{r=1}^k \|\boldsymbol{v}_r\|_q^p \right)^{\frac{1}{p}}$. The number of non-zero rows of $\boldsymbol{V}$ is denoted by $\|\boldsymbol{V}\|_{0,\infty}$.

**Factorization machines (FMs).** Given an input vector $\boldsymbol{x} \in \mathbb{R}^d$, FMs predict a scalar output by

$$\hat{y}_{\text{FM}} := \boldsymbol{w}^{\text{T}} \boldsymbol{x} + \sum_{i<j} w_{i,j} x_i x_j,$$

where $\boldsymbol{w} \in \mathbb{R}^d$ contains feature weights and $\boldsymbol{W} \in \mathbb{R}^{d \times d}$ is a low-rank matrix that contains pairwise feature interaction weights. To obtain a low-rank $\boldsymbol{W}$, [25] originally proposed to use a change of variable $\boldsymbol{W} = \boldsymbol{H}^{\text{T}} \boldsymbol{H}$, where $\boldsymbol{H} \in \mathbb{R}^{k \times d}$ (with $k \in \mathbb{N}_+$ a rank parameter) and to learn $\boldsymbol{H}$ instead. Noting that this quadratic model results in a non-convex problem in $\boldsymbol{H}$, [4, 31] proposed to convexify the problem by learning $\boldsymbol{W}$ directly but to encourage low rank using a nuclear norm on $\boldsymbol{W}$. For learning, [4] proposed a conditional gradient like approach with global convergence guarantees.

**Polynomial networks (PNs).** PNs are a recently-proposed form of neural network where the usual activation function is replaced with a squared activation. Formally, PNs predict a scalar output by

$$\hat{y}_{\text{PN}} := \boldsymbol{w}^{\text{T}} \boldsymbol{x} + \boldsymbol{v}^{\text{T}} \sigma(\boldsymbol{H}\boldsymbol{x}) = \boldsymbol{w}^{\text{T}} \boldsymbol{x} + \sum_{r=1}^k v_r \ \sigma(\boldsymbol{h}_r^{\text{T}} \boldsymbol{x}),$$

where $\sigma(a) = a^2$ (evaluated element-wise) is the squared activation, $\boldsymbol{v} \in \mathbb{R}^k$ is the output layer vector, $\boldsymbol{H} \in \mathbb{R}^{k \times d}$ is the hidden layer matrix and $k$ is the number of hidden units. Because the

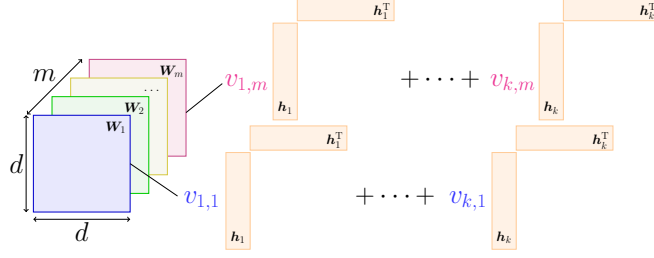

Figure 1: Our multi-output PNs / FMs learn a tensor whose slices **share a common basis** $\{\boldsymbol{h}_r\}_{r=1}^k$.

r.h.s term can be rewritten as $\boldsymbol{x}^{\mathrm{T}}\boldsymbol{W}\boldsymbol{x} = \sum_{i,j=1}^d w_{i,j}x_i x_j$ if we set $\boldsymbol{W} = \boldsymbol{H}^{\mathrm{T}}\mathrm{diag}(\boldsymbol{v})\boldsymbol{H}$, we see that PNs are clearly a slight variation of FMs and that learning $(\boldsymbol{v}, \boldsymbol{H})$ can be recast as learning a low-rank matrix $\boldsymbol{W}$. Based on this observation, [20] proposed to use GECO [26], a greedy algorithm for convex optimization with a low-rank constraint, similar to the conditional gradient algorithm. [13] proposed a learning algorithm for PNs with global optimality guarantees but their theory imposes non-negativity on the network parameters and they need one distinct hyper-parameter per hidden unit to avoid trivial models. Other low-rank polynomial models were recently introduced in [29, 23] but using a tensor network (a.k.a. tensor train) instead of the canonical polyadic (CP) decomposition.

## 3 A convex formulation of multi-output PNs and FMs

In this section, we generalize PNs and FMs to multi-output problems. For the sake of concreteness, we focus on PNs for multi-class classification. The extension to FMs is straightforward and simply requires to replace $\sigma(\boldsymbol{h}^{\mathrm{T}}\boldsymbol{x}) = (\boldsymbol{h}^{\mathrm{T}}\boldsymbol{x})^2$ by $\sigma_{\mathrm{ANOVA}}(\boldsymbol{h}, \boldsymbol{x}) := \sum_{i<j} x_i h_i x_j h_j$, as noted in [5].

The predictions of multi-class PNs can be naturally defined as $\hat{y}_{\mathrm{MPN}} := \mathrm{argmax}_{c\in[m]} \boldsymbol{w}_c^{\mathrm{T}}\boldsymbol{x} + \boldsymbol{x}^{\mathrm{T}}\boldsymbol{W}_c\boldsymbol{x}$, where $m$ is the number of classes, $\boldsymbol{w}_c \in \mathbb{R}^d$ and $\boldsymbol{W}_c \in \mathbb{R}^{d\times d}$ is low-rank. Following [5], we can model the linear term directly in the quadratic term if we augment all data points with an extra feature of value 1, i.e., $\boldsymbol{x}^{\mathrm{T}} \leftarrow [1, \boldsymbol{x}^{\mathrm{T}}]$. We will therefore simply assume $\hat{y}_{\mathrm{MPN}} = \mathrm{argmax}_{c\in[m]} \boldsymbol{x}^{\mathrm{T}}\boldsymbol{W}_c\boldsymbol{x}$ henceforth. Our main proposal in this paper is to decompose $\boldsymbol{W}_1, \ldots, \boldsymbol{W}_m$ using a **shared** basis:

$$\boldsymbol{W}_c = \boldsymbol{H}^{\mathrm{T}}\mathrm{diag}(\boldsymbol{v}_{:,c})\boldsymbol{H} = \sum_{r=1}^k v_{r,c}\boldsymbol{h}_r\boldsymbol{h}_r^{\mathrm{T}} \quad \forall c\in[m], \tag{1}$$

where, in neural network terminology, $\boldsymbol{H} \in \mathbb{R}^{k\times d}$ can be interpreted as a hidden layer matrix and $\boldsymbol{V} \in \mathbb{R}^{k\times m}$ as an output layer matrix. Compared to the naive approach of decomposing each $\boldsymbol{W}_c$ as $\boldsymbol{W}_c = \boldsymbol{H}_c^{\mathrm{T}}\mathrm{diag}(\boldsymbol{v}_{:,c})\boldsymbol{H}_c$, this reduces the number of parameters from $m(dk+k)$ to $dk+mk$.

While a nuclear norm could be used to promote a low rank on each $\boldsymbol{W}_c$, similarly as in [4, 31], this is clearly not sufficient to impose a shared basis. A naive approach would be to use non-orthogonal joint diagonalization as a post-processing. However, because this is a non-convex problem for which no globally convergent algorithm is known [24], this would result in a loss of accuracy. Our key idea is to cast the problem of learning a multi-output PN as that of learning an **infinite but row-wise sparse matrix**. Without loss of generality, we assume that basis vectors (hidden units) lie in the unit ball. We therefore denote the set of basis vectors by $\mathcal{H} := \{\boldsymbol{h} \in \mathbb{R}^d : \|\boldsymbol{h}\|_2 \le 1\}$. Let us denote this infinite matrix by $\boldsymbol{U} \in \mathbb{R}^{|\mathcal{H}|\times m}$ (we use a discrete notation for simplicity). We can then write

$$\hat{y}_{\mathrm{MPN}} = \mathrm{argmax}_{c\in[m]} \boldsymbol{o}(\boldsymbol{x}; \boldsymbol{U})_c \quad \text{where} \quad \boldsymbol{o}(\boldsymbol{x}; \boldsymbol{U}) := \sum_{\boldsymbol{h}\in\mathcal{H}} \sigma(\boldsymbol{h}^{\mathrm{T}}\boldsymbol{x})\boldsymbol{u}_{\boldsymbol{h}} \in \mathbb{R}^m \quad \text{and}$$

$\boldsymbol{u}_{\boldsymbol{h}} \in \mathbb{R}^m$ denotes the weights of basis $\boldsymbol{h}$ across all classes (outputs). In this formulation, we have $\boldsymbol{W}_c = \sum_{\boldsymbol{h}\in\mathcal{H}} u_{\boldsymbol{h},c}\boldsymbol{h}\boldsymbol{h}^{\mathrm{T}}$ and sharing a common basis (hidden units) amounts to encouraging the rows of $\boldsymbol{U}$, $\boldsymbol{u}_{\boldsymbol{h}}$, to be either dense or entirely sparse. This can be naturally achieved using group-sparsity inducing penalties. Intuitively, $\boldsymbol{V}$ in (1) can be thought as $\boldsymbol{U}$ restricted to its row support. Define the training set by $\boldsymbol{X} \in \mathbb{R}^{n\times d}$ and $\boldsymbol{y} \in [m]^n$. We then propose to solve the convex problem

$$\min_{\Omega(\boldsymbol{U})\le\tau} F(\boldsymbol{U}) := \sum_{i=1}^n \ell\left(y_i, \boldsymbol{o}(\boldsymbol{x}_i; \boldsymbol{U})\right), \tag{2}$$

Table 1: Sparsity-inducing penalties considered in this paper. With some abuse of notation, we denote by $\boldsymbol{e_h}$ and $\boldsymbol{e_c}$ standard basis vectors of dimension $|\mathcal{H}|$ and $m$, respectively. Selecting an optimal basis vector $\boldsymbol{h}^\star$ to add is a non-convex optimization problem. The constant $\epsilon \in (0,1)$ is the tolerance parameter used for the power method and $\nu$ is the multiplicative approximation we guarantee.

| | $\Omega(\boldsymbol{U})$ | $\Omega^*(\boldsymbol{G})$ | $\boldsymbol{\Delta}^\star \in \tau \cdot \partial\Omega^*(\boldsymbol{G})$ | Subproblem | $\nu$ |
|---|---|---|---|---|---|
| $l_1$ (lasso) | $\|\boldsymbol{U}\|_1$ | $\|\boldsymbol{G}\|_\infty$ | $\tau \operatorname{sign}(g_{\boldsymbol{h}^\star,c^\star})\boldsymbol{e}_{\boldsymbol{h}^\star}\boldsymbol{e}_{c^\star}^{\mathrm{T}}$ | $\boldsymbol{h}^\star, c^\star \in \underset{\boldsymbol{h}\in\mathcal{H},c\in[m]}{\operatorname{argmax}} |g_{\boldsymbol{h},c}|$ | $1-\epsilon$ |
| $l_1/l_2$ (group lasso) | $\|\boldsymbol{U}\|_{1,2}$ | $\|\boldsymbol{G}\|_{\infty,2}$ | $\tau\boldsymbol{e}_{\boldsymbol{h}^\star}\boldsymbol{g}_{\boldsymbol{h}^\star}^{\mathrm{T}}/\|\boldsymbol{g}_{\boldsymbol{h}^\star}\|_2$ | $\boldsymbol{h}^\star \in \underset{\boldsymbol{h}\in\mathcal{H}}{\operatorname{argmax}}\|\boldsymbol{g}_{\boldsymbol{h}}\|_2$ | $\frac{1-\epsilon}{\sqrt{m}}$ |
| $l_1/l_\infty$ | $\|\boldsymbol{U}\|_{1,\infty}$ | $\|\boldsymbol{G}\|_{\infty,1}$ | $\tau\boldsymbol{e}_{\boldsymbol{h}^\star}\operatorname{sign}(\boldsymbol{g}_{\boldsymbol{h}^\star})^{\mathrm{T}}$ | $\boldsymbol{h}^\star \in \underset{\boldsymbol{h}\in\mathcal{H}}{\operatorname{argmax}}\|\boldsymbol{g}_{\boldsymbol{h}}\|_1$ | $\frac{1-\epsilon}{m}$ |

where $\ell$ is a smooth and convex multi-class loss function (cf. Appendix A for three common examples), $\Omega$ is a sparsity-inducing penalty and $\tau > 0$ is a hyper-parameter. In this paper, we focus on the $l_1$ (lasso), $l_1/l_2$ (group lasso) and $l_1/l_\infty$ penalties for $\Omega$, cf. Table 1. However, as we shall see, solving (2) is more challenging with the $l_1/l_2$ and $l_1/l_\infty$ penalties than with the $l_1$ penalty. Although our formulation is based on an infinite view, we next show that $\boldsymbol{U}^\star$ has finite row support.

**Proposition 1** *Finite row support of $\boldsymbol{U}^\star$ for multi-output PNs and FMs*

*Let $\boldsymbol{U}^\star$ be an optimal solution of* (2)*, where $\Omega$ is one of the penalties in Table 1. Then,*
$\|\boldsymbol{U}^\star\|_{0,\infty} \leq nm + 1$. *If $\Omega(\cdot) = \|\cdot\|_1$, we can tighten this bound to $\|\boldsymbol{U}^\star\|_{0,\infty} \leq \min(nm+1, dm)$.*

Proof is in Appendix B.1. It is open whether we can tighten this result when $\Omega = \|\cdot\|_{1,2}$ or $\|\cdot\|_{1,\infty}$.

## 4 A conditional gradient algorithm with approximate basis vector selection

At first glance, learning with an infinite number of basis vectors seems impossible. In this section, we show how the well-known conditional gradient algorithm [11, 15] combined with group-sparsity inducing penalties naturally leads to a greedy algorithm that **selects and adds basis vectors that are useful across all outputs**. On every iteration, the conditional gradient algorithm performs updates of the form $\boldsymbol{U}^{(t+1)} = (1-\gamma)\boldsymbol{U}^{(t)} + \gamma\boldsymbol{\Delta}^\star$, where $\gamma \in [0,1]$ is a step size and $\boldsymbol{\Delta}^\star$ is obtained by solving a linear approximation of the objective around the current iterate $\boldsymbol{U}^{(t)}$:

$$\boldsymbol{\Delta}^\star \in \underset{\Omega(\boldsymbol{\Delta})\leq\tau}{\operatorname{argmin}}\langle\boldsymbol{\Delta}, \nabla F(\boldsymbol{U}^{(t)})\rangle = \tau \cdot \underset{\Omega(\boldsymbol{\Delta})\leq 1}{\operatorname{argmax}}\langle\boldsymbol{\Delta}, -\nabla F(\boldsymbol{U}^{(t)})\rangle. \qquad (3)$$

Let us denote the negative gradient $-\nabla F(\boldsymbol{U})$ by $\boldsymbol{G} \in \mathbb{R}^{|\mathcal{H}|\times m}$ for short. Its elements are defined by

$$g_{\boldsymbol{h},c} = -\sum_{i=1}^n \sigma(\boldsymbol{h}^{\mathrm{T}}\boldsymbol{x}_i)\nabla\ell\left(y_i, \boldsymbol{o}(\boldsymbol{x}_i;\boldsymbol{U})\right)_c,$$

where $\nabla\ell(y,\boldsymbol{o}) \in \mathbb{R}^m$ is the gradient of $\ell$ w.r.t. $\boldsymbol{o}$ (cf. Appendix A). For ReLu activations, solving (3) is known to be NP-hard [1]. Here, we focus on quadratic activations, for which we will be able to provide approximation guarantees. Plugging the expression of $\sigma$, we get

$$g_{\boldsymbol{h},c} = -\boldsymbol{h}^{\mathrm{T}}\boldsymbol{\Gamma}_c\boldsymbol{h} \text{ where } \boldsymbol{\Gamma}_c := \boldsymbol{X}^{\mathrm{T}}\boldsymbol{D}_c\boldsymbol{X} \text{ (PN) or } \boldsymbol{\Gamma}_c := \frac{1}{2}\left(\boldsymbol{X}^{\mathrm{T}}\boldsymbol{D}_c\boldsymbol{X} - \boldsymbol{D}_c\sum_{i=1}^n \operatorname{diag}(\boldsymbol{x}_i)^2\right) \text{ (FM)}$$

and $\boldsymbol{D}_c \in \mathbb{R}^{n\times n}$ is a diagonal matrix such that $(\boldsymbol{D}_c)_{i,i} := \nabla\ell(y_i, \boldsymbol{o}(\boldsymbol{x}_i;\boldsymbol{U}))_c$. Let us recall the definition of the dual norm of $\Omega$: $\Omega^*(\boldsymbol{G}) := \max_{\Omega(\boldsymbol{\Delta})\leq 1}\langle\boldsymbol{\Delta}, \boldsymbol{G}\rangle$. By comparing this equation to (3), we see that $\boldsymbol{\Delta}^\star$ is the argument that achieves the maximum in the dual norm $\Omega^*(\boldsymbol{G})$, up to a constant factor $\tau$. It is easy to verify that any element in the subdifferential of $\Omega^*(\boldsymbol{G})$, which we denote by $\partial\Omega^*(\boldsymbol{G}) \subseteq \mathbb{R}^{|\mathcal{H}|\times m}$, achieves that maximum, i.e., $\boldsymbol{\Delta}^\star \in \tau \cdot \partial\Omega^*(\boldsymbol{G})$.

**Basis selection.** As shown in Table 1, elements of $\partial\Omega^*(\boldsymbol{G})$ (subgradients) are $|\mathcal{H}| \times m$ matrices with a **single non-zero row** indexed by $\boldsymbol{h}^\star$, where $\boldsymbol{h}^\star$ is an optimal basis (hidden unit) selected by

$$\boldsymbol{h}^\star \in \underset{\boldsymbol{h}\in\mathcal{H}}{\operatorname{argmax}}\|\boldsymbol{g}_{\boldsymbol{h}}\|_p, \qquad (4)$$

and where $p = \infty$ when $\Omega = \|\cdot\|_1$, $p = 2$ when $\Omega = \|.\|_{1,2}$ and $p = 1$ when $\Omega = \|\cdot\|_{1,\infty}$. We call (4) a basis vector selection criterion. Although this selection criterion was derived from the linearization of the objective, it is fairly natural: it chooses the basis vector with largest "violation", as measured by the $l_p$ norm of the negative gradient row $\boldsymbol{g_h}$.

**Multiplicative approximations.** The key challenge in solving (3) or equivalently (4) arises from the fact that $\boldsymbol{G}$ has infinitely many rows $\boldsymbol{g_h}$. We therefore cast basis vector selection as a continuous optimization problem w.r.t. $\boldsymbol{h}$. Surprisingly, although the entire objective (2) is convex, (4) is not. Instead of the exact maximum, we will therefore only require to find a $\hat{\boldsymbol{\Delta}} \in \mathbb{R}^{|\mathcal{H}| \times m}$ that satisfies

$$\Omega(\hat{\boldsymbol{\Delta}}) \le \tau \quad \text{and} \quad \langle \hat{\boldsymbol{\Delta}}, \boldsymbol{G} \rangle \ge \nu \langle \boldsymbol{\Delta}^\star, \boldsymbol{G} \rangle,$$

where $\nu \in (0, 1]$ is a multiplicative approximation (higher is better). It is easy to verify that this is equivalent to replacing the optimal $\boldsymbol{h}^\star$ by an approximate $\hat{\boldsymbol{h}} \in \mathcal{H}$ that satisfies $\|\boldsymbol{g_{\hat{h}}}\|_p \ge \nu \|\boldsymbol{g_{h^\star}}\|_p$.

**Sparse case.** When $\Omega(\cdot) = \|\cdot\|_1$, we need to solve

$$\max_{\boldsymbol{h} \in \mathcal{H}} \|\boldsymbol{g_h}\|_\infty = \max_{\boldsymbol{h} \in \mathcal{H}} \max_{c \in [m]} |\boldsymbol{h}^\mathrm{T} \boldsymbol{\Gamma}_c \boldsymbol{h}| = \max_{c \in [m]} \max_{\boldsymbol{h} \in \mathcal{H}} |\boldsymbol{h}^\mathrm{T} \boldsymbol{\Gamma}_c \boldsymbol{h}|.$$

It is well known that the optimal solution of $\max_{\boldsymbol{h} \in \mathcal{H}} |\boldsymbol{h}^\mathrm{T} \boldsymbol{\Gamma}_c \boldsymbol{h}|$ is the dominant eigenvector of $\boldsymbol{\Gamma}_c$. Therefore, we simply need to find the dominant eigenvector $\boldsymbol{h}_c$ of each $\boldsymbol{\Gamma}_c$ and select $\hat{\boldsymbol{h}}$ as the $\boldsymbol{h}_c$ with largest singular value $|\boldsymbol{h}_c^\mathrm{T} \boldsymbol{\Gamma}_c \boldsymbol{h}_c|$. Using the power method, we can find an $\boldsymbol{h}_c$ that satisfies

$$|\boldsymbol{h}_c^\mathrm{T} \boldsymbol{\Gamma}_c \boldsymbol{h}_c| \ge (1 - \epsilon) \max_{\boldsymbol{h} \in \mathcal{H}} |\boldsymbol{h}^\mathrm{T} \boldsymbol{\Gamma}_c \boldsymbol{h}|, \tag{5}$$

for some tolerance parameter $\epsilon \in (0, 1)$. The procedure takes $\mathcal{O}(N_c \log(d)/\epsilon)$ time, where $N_c$ is the number of non-zero elements in $\boldsymbol{\Gamma}_c$ [26]. Taking the maximum w.r.t. $c \in [m]$ on both sides of (5) leads to $\|\boldsymbol{g_{\hat{h}}}\|_\infty \ge \nu \|\boldsymbol{g_{h^\star}}\|_\infty$, where $\nu = 1 - \epsilon$. However, using $\Omega = \|\cdot\|_1$ does not encourage selecting an $\hat{\boldsymbol{h}}$ that is useful for all outputs. In fact, when $\Omega = \|\cdot\|_1$, our approach is equivalent to imposing independent nuclear norms on $\boldsymbol{W}_1, \ldots, \boldsymbol{W}_m$.

**Group-sparse cases.** When $\Omega(\cdot) = \|.\|_{1,2}$ or $\Omega(\cdot) = \|.\|_{1,\infty}$, we need to solve

$$\max_{\boldsymbol{h} \in \mathcal{H}} \|\boldsymbol{g_h}\|_2^2 = \max_{\boldsymbol{h} \in \mathcal{H}} f_2(\boldsymbol{h}) := \sum_{c=1}^m (\boldsymbol{h}^\mathrm{T} \boldsymbol{\Gamma}_c \boldsymbol{h})^2 \quad \text{or} \quad \max_{\boldsymbol{h} \in \mathcal{H}} \|\boldsymbol{g_h}\|_1 = \max_{\boldsymbol{h} \in \mathcal{H}} f_1(\boldsymbol{h}) := \sum_{c=1}^m |\boldsymbol{h}^\mathrm{T} \boldsymbol{\Gamma}_c \boldsymbol{h}|,$$

respectively. Unlike the $l_1$-constrained case, we are clearly selecting a basis vector with largest violation **across all outputs**. However, we are now faced with a more difficult non-convex optimization problem. Our strategy is to first choose an initialization $\boldsymbol{h}^{(0)}$ which guarantees a certain multiplicative approximation $\nu$, then refine the solution using a monotonically non-increasing iterative procedure.

*Initialization.* We simply choose $\boldsymbol{h}^{(0)}$ as the approximate solution of the $\Omega = \|\cdot\|_1$ case, i.e., we have

$$\|\boldsymbol{g_{h^{(0)}}}\|_\infty \ge (1 - \epsilon) \max_{\boldsymbol{h} \in \mathcal{H}} \|\boldsymbol{g_h}\|_\infty.$$

Now, using $\sqrt{m}\|\boldsymbol{x}\|_\infty \ge \|\boldsymbol{x}\|_2 \ge \|\boldsymbol{x}\|_\infty$ and $m\|\boldsymbol{x}\|_\infty \ge \|\boldsymbol{x}\|_1 \ge \|\boldsymbol{x}\|_\infty$, this immediately implies

$$\|\boldsymbol{g_{h^{(0)}}}\|_p \ge \nu \max_{\boldsymbol{h} \in \mathcal{H}} \|\boldsymbol{g_h}\|_p,$$

with $\nu = \frac{1-\epsilon}{\sqrt{m}}$ if $p = 2$ and $\nu = \frac{1-\epsilon}{m}$ if $p = 1$.

*Refining the solution.* We now apply another instance of the conditional gradient algorithm to solve the subproblem $\max_{\|\boldsymbol{h}\|_2 \le 1} f_p(\boldsymbol{h})$ itself, leading to the following iterates:

$$\boldsymbol{h}^{(t+1)} = (1 - \eta_t)\boldsymbol{h}^{(t)} + \eta_t \frac{\nabla f_p(\boldsymbol{h}^{(t)})}{\|\nabla f_p(\boldsymbol{h}^{(t)})\|_2}, \tag{6}$$

where $\eta_t \in [0, 1]$. Following [3, Section 2.2.2], if we use the Armijo rule to select $\eta_t$, every limit point of the sequence $\{\boldsymbol{h}^{(t)}\}$ is a stationary point of $f_p$. In practice, we observe that $\eta_t = 1$ is almost always selected. Note that when $\eta_t = 1$ and $m = 1$ (i.e., single-output case), our refining algorithm **recovers the power method**. Generalized power methods were also studied for structured matrix factorization [16, 21], but with different objectives and constraints. Since the conditional gradient

---
**Algorithm 1** Multi-output PN/FM training
---
**Input:** $\boldsymbol{X}$, $\boldsymbol{y}$, $k$, $\tau$
$\boldsymbol{H} \leftarrow [\,]$, $\boldsymbol{V} \leftarrow [\,]$
**for** $t := 1, \dots, k$ **do**
    Compute $\boldsymbol{o}_i := \sum_{r=1}^{t-1} \sigma(\boldsymbol{h}_r^{\mathrm{T}} \boldsymbol{x}_i) \boldsymbol{v}_r \ \forall i \in [n]$
    Let $\boldsymbol{g_h} := [-\boldsymbol{h}^{\mathrm{T}} \boldsymbol{\Gamma}_1 \boldsymbol{h}, \dots, -\boldsymbol{h}^{\mathrm{T}} \boldsymbol{\Gamma}_m \boldsymbol{h}]^{\mathrm{T}}$
    Find $\hat{\boldsymbol{h}} \approx \operatorname{argmax}_{\boldsymbol{h} \in \mathcal{H}} \|\boldsymbol{g_h}\|_p$
    Append $\hat{\boldsymbol{h}}$ to $\boldsymbol{H}$ and $\boldsymbol{0}$ to $\boldsymbol{V}$
    $\boldsymbol{V} \leftarrow \underset{\Omega(\boldsymbol{V}) \leq \tau}{\operatorname{argmin}} \ F_t(\boldsymbol{V}, \boldsymbol{H})$
    Optional: $\boldsymbol{V}, \boldsymbol{H} \leftarrow \underset{\substack{\Omega(\boldsymbol{V}) \leq \tau \\ \boldsymbol{h}_r \in \mathcal{H} \ \forall r \in [t]}}{\operatorname{argmin}} \ F_t(\boldsymbol{V}, \boldsymbol{H})$
**end for**
**Output:** $\boldsymbol{V}$, $\boldsymbol{H}$ (equivalent to $\boldsymbol{U} = \sum_{t=1}^{k} \boldsymbol{e}_{\boldsymbol{h}_t} \boldsymbol{v}_t^{\mathrm{T}}$)
---

algorithm assumes a differentiable function, in the case $p = 1$, we replace the absolute function with the Huber function $|x| \approx \frac{1}{2} x^2$ if $|x| \leq 1$, $|x| - \frac{1}{2}$ otherwise.

**Corrective refitting step.** After $t$ iterations, $\boldsymbol{U}^{(t)}$ contains at most $t$ non-zero rows. We can therefore always store $\boldsymbol{U}^{(t)}$ as $\boldsymbol{V}^{(t)} \in \mathbb{R}^{t \times m}$ (the output layer matrix) and $\boldsymbol{H}^{(t)} \in \mathbb{R}^{t \times d}$ (the basis vectors / hidden units added so far). In order to improve accuracy, on iteration $t$, we can then refit the objective $F_t(\boldsymbol{V}, \boldsymbol{H}) := \sum_{i=1}^{n} \ell \left( y_i, \sum_{r=1}^{t} \sigma(\boldsymbol{h}_r^{\mathrm{T}} \boldsymbol{x}_i) \boldsymbol{v}_r \right)$. We consider two kinds of corrective steps, a convex one that minimizes $F_t(\boldsymbol{V}, \boldsymbol{H}^{(t)})$ w.r.t. $\boldsymbol{V} \in \mathbb{R}^{t \times m}$ and an optional non-convex one that minimizes $F_t(\boldsymbol{V}, \boldsymbol{H})$ w.r.t. both $\boldsymbol{V} \in \mathbb{R}^{t \times m}$ and $\boldsymbol{H} \in \mathbb{R}^{t \times d}$. Refitting allows to remove previously-added bad basis vectors, thanks to the use of sparsity-inducing penalties. Similar refitting procedures are commonly used in matching pursuit [22]. The entire procedure is summarized in Algorithm 1 and implementation details are given in Appendix D.

## 5   Analysis of Algorithm 1

The main difficulty in analyzing the convergence of Algorithm 1 stems from the fact that we cannot solve the basis vector selection subproblem globally when $\Omega = \|\cdot\|_{1,2}$ or $\|\cdot\|_{1,\infty}$. Therefore, we need to develop an analysis that can cope with the multiplicative approximation $\nu$. Multiplicative approximations were also considered in [18] but the condition they require is too stringent (cf. Appendix B.2 for a detailed discussion). The next theorem guarantees the number of iterations needed to output a multi-output network that achieves as small objective value as an optimal solution of (2).

**Theorem 1** *Convergence of Algorithm 1*

*Assume $F$ is smooth with constant $\beta$. Let $\boldsymbol{U}^{(t)}$ be the output after $t$ iterations of Algorithm 1 run with constraint parameter $\frac{\tau}{\nu}$. Then, $F(\boldsymbol{U}^{(t)}) - \min_{\Omega(\boldsymbol{U}) \leq \tau} F(\boldsymbol{U}) \leq \epsilon \ \forall t \geq \dfrac{8 \tau^2 \beta}{\epsilon \nu^2} - 2.$*

In [20], single-output PNs were trained using GECO [26], a greedy algorithm with similar $\mathcal{O}\!\left(\frac{\tau^2 \beta}{\epsilon \nu^2}\right)$ guarantees. However, GECO is limited to learning infinite vectors (not matrices) and it does not constrain its iterates like we do. Hence GECO cannot remove bad basis vectors. The proof of Theorem 1 and a detailed comparison with GECO are given in Appendix B.2. Finally, we note that the infinite dimensional view is also key to convex neural networks [2, 1]. However, to our knowledge, we are the first to give an explicit multiplicative approximation guarantee for a non-linear multi-output network.

## 6   Experimental results

### 6.1   Experimental setup

**Datasets.** For our multi-class experiments, we use four publicly-available datasets: segment (7 classes), vowel (11 classes), satimage (6 classes) and letter (26 classes) [12]. Quadratic models sub-

stantially improve over linear models on these datasets. For our recommendation system experiments, we use the MovieLens 100k and 1M datasets [14]. See Appendix E for complete details.

**Model validation.** The greedy nature of Algorithm 1 allows us to easily interleave training with model validation. Concretely, we use an outer loop (embarrassingly parallel) for iterating over the range of possible regularization parameters, and an inner loop (Algorithm 1, sequential) for increasing the number of basis vectors. Throughout our experiments, we use 50% of the data for training, 25% for validation, and 25% for evaluation. Unless otherwise specified, we use a multi-class logistic loss.

## 6.2 Method comparison for the basis vector (hidden unit) selection subproblem

As we mentioned previously, the linearized subproblem (basis vector selection) for the $l_1/l_2$ and $l_1/l_\infty$ constrained cases involves a significantly more challenging non-convex optimization problem. In this section, we compare different methods for obtaining an approximate solution $\hat{h}$ to (4). We focus on the $\ell_1/\ell_\infty$ case, since we have a method for computing the true global solution $h^\star$, albeit with exponential complexity in $m$ (cf. Appendix C). This allows us to report the empirically observed multiplicative approximation factor $\hat{\nu} := f_1(\hat{h})/f_1(h^\star)$.

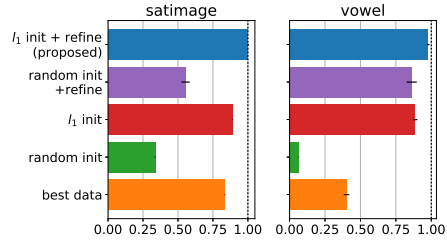

Figure 2: Empirically observed multiplicative approximation factor $\hat{\nu} = f_1(\hat{h})/f_1(h^\star)$.

**Compared methods.** We compare $l_1$ *init + refine* (proposed), *random init + refine*, $l_1$ *init* (without refine), *random init* and *best data*: $\hat{h} = x_{i^\star}/\|x_{i^\star}\|_2$ where $i^\star = \underset{i \in [n]}{\mathrm{argmax}} \, f_1(x_i/\|x_i\|_2)$.

**Results.** We report $\hat{\nu}$ in Figure 2. $l_1$ *init + refine* achieves nearly the global maximum on both datasets and outperforms *random init + refine*, showing the effectiveness of the proposed initialization and that the iterative update (6) can get stuck in a bad local minimum if initialized badly. On the other hand, $l_1$ *init + refine* outperforms $l_1$ *init* alone, showing the importance of iteratively refining the solution. *Best data*, a heuristic similar to that of approximate kernel SVMs [7], is not competitive.

## 6.3 Sparsity-inducing penalty comparison

In this section, we compare the $l_1$, $l_1/l_2$ and $l_1/l_\infty$ penalties for the choice of $\Omega$, when varying the maximum number of basis vectors (hidden units). Figure 3 indicates test set accuracy when using output layer refitting. We also include linear logistic regression, kernel SVMs and the Nyström method as baselines. For the latter two, we use the quadratic kernel $(x_i^\mathrm{T} x_j + 1)^2$. Hyper-parameters are chosen so as to maximize validation set accuracy.

**Results.** On the vowel (11 classes) and letter (26 classes) datasets, $l_1/l_2$ and $l_1/l_\infty$ penalties outperform $l_1$ norm starting from 20 and 75 hidden units, respectively. On satimage (6 classes) and segment (7 classes), we observed that the three penalties are mostly similar (not shown). We hypothesize that $l_1/l_2$ and $l_1/l_\infty$ penalties make a bigger difference when the number of classes is large. Multi-output PNs substantially outperform the Nyström method with comparable number of basis vectors (hidden units). Multi-output PNs reach the same test accuracy as kernel SVMs with very few basis vectors on vowel and satimage but appear to require at least 100 basis vectors to reach good performance on letter. This is not surprising, since kernel SVMs require 3,208 support vectors on letter, as indicated in Table 2 below.

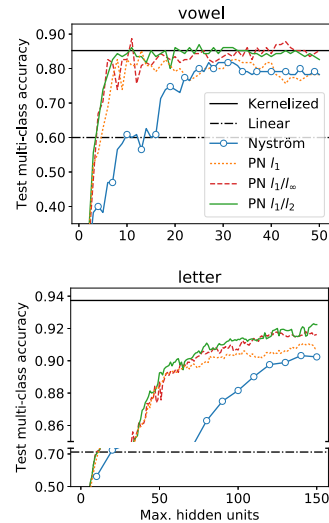

Figure 3: Penalty comparison.

## 6.4 Multi-class benchmark comparison

**Compared methods.** We compare the proposed conditional gradient algorithm with output layer refitting only and with both output and hidden layer refitting; projected gradient descent (FISTA)

Table 2: Muli-class test accuracy and number of basis vectors / support vectors.

| | segment | vowel | satimage | letter |
|---|---|---|---|---|
| **Conditional gradient** (full refitting, proposed) | | | | |
| $l_1$ | 96.71 (41) | 87.83 (12) | 89.80 (25) | 92.29 (150) |
| $l_1/l_2$ | 96.71 (40) | 89.57 (15) | 89.08 (18) | 91.81 (106) |
| $l_1/l_\infty$ | 96.71 (24) | 86.96 (15) | 88.99 (20) | 92.35 (149) |
| **Conditional gradient** (output-layer refitting, proposed) | | | | |
| $l_1$ | 97.05 (20) | 80.00 (21) | 89.71 (40) | 91.01 (139) |
| $l_1/l_2$ | 96.36 (21) | 85.22 (15) | 89.71 (50) | 92.24 (150) |
| $l_1/l_\infty$ | 96.19 (16) | 86.96 (41) | 89.35 (41) | 91.68 (128) |
| **Projected gradient descent** (random init) | | | | |
| $l_1$ | 96.88 (50) | 79.13 (50) | 89.53 (50) | 88.45 (150) |
| $l_1/l_2$ | 96.88 (50) | 80.00 (48) | 89.80 (50) | 88.45 (150) |
| $l_1/l_\infty$ | 96.71 (50) | 83.48 (50) | 89.08 (50) | 88.45 (150) |
| $l_2^2$ | 96.88 (50) | 81.74 (50) | 89.98 (50) | 88.45 (150) |
| **Baselines** | | | | |
| Linear | 92.55 | 60.00 | 83.03 | 71.17 |
| Kernelized | 96.71 (238) | 85.22 (189) | 89.53 (688) | 93.73 (3208) |
| OvR PN | 94.63 | 73.91 | 89.44 | 75.36 |

with random initialization; linear and kernelized models; one-vs-rest PNs (i.e., fit one PN per class). We focus on PNs rather than FMs since they are known to work better on classification tasks [5].

**Results** are included in Table 2. From these results, we can make the following observations and conclusions. When using output-layer refitting on vowel and letter (two datasets with more than 10 classes), group-sparsity inducing penalties lead to better test accuracy. This is to be expected, since these penalties select basis vectors that are useful across all classes. When using full hidden layer and output layer refitting, $l_1$ catches up with $l_1/l_2$ and $l_1/l_\infty$ on the vowel and letter datasets. Intuitively, the basis vector selection becomes less important if we make more effort at every iteration by refitting the basis vectors themselves. However, on vowel, $l_1/l_2$ is still substantially better than $l_1$ (89.57 vs. 87.83).

Compared to projected gradient descent with random initialization, our algorithm (for both output and full refitting) is better on ¾ ($l_1$), 2/4 ($l_1/l_2$) and ¾ ($l_1/l_\infty$) of the datasets. In addition, with our algorithm, the best model (chosen against the validation set) is substantially sparser. Multi-output PNs substantially outperform OvR PNs. This is to be expected, since multi-output PNs learn to share basis vectors across different classes.

## 6.5   Recommender system experiments using ordinal regression

A straightforward way to implement recommender systems consists in training a single-output model to regress ratings from one-hot encoded user and item indices [25]. Instead of a single-output PN or FM, we propose to use ordinal McRank, a **reduction from ordinal regression to multi-output binary classification**, which is known to achieve good nDCG (normalized discounted cumulative gain) scores [19]. This reduction involves training a probabilistic binary classifier for each of the $m$ relevance levels (for instance, $m = 5$ in the MovieLens datasets). The expected relevance of $\boldsymbol{x}$ (e.g. the concatenation of the one-hot encoded user and item indices) is then computed by

$$\hat{y} = \sum_{c=1}^{m} c\, p(y = c \mid \boldsymbol{x}) = \sum_{c=1}^{m} c \Big[ p(y \leq c \mid \boldsymbol{x}) - p(y \leq c - 1 \mid \boldsymbol{x}) \Big],$$

where we use the convention $p(y \leq 0 \mid \boldsymbol{x}) = 0$. Thus, all we need to do to use ordinal McRank is to train a probabilistic binary classifier $p(y \leq c \mid \boldsymbol{x})$ for all $c \in [m]$.

Our key proposal is to use a multi-output model to learn all $m$ classifiers simultaneously, i.e., in a multi-task fashion. Let $\boldsymbol{x}_i$ be a vector representing a user-item pair with corresponding rating $y_i$, for

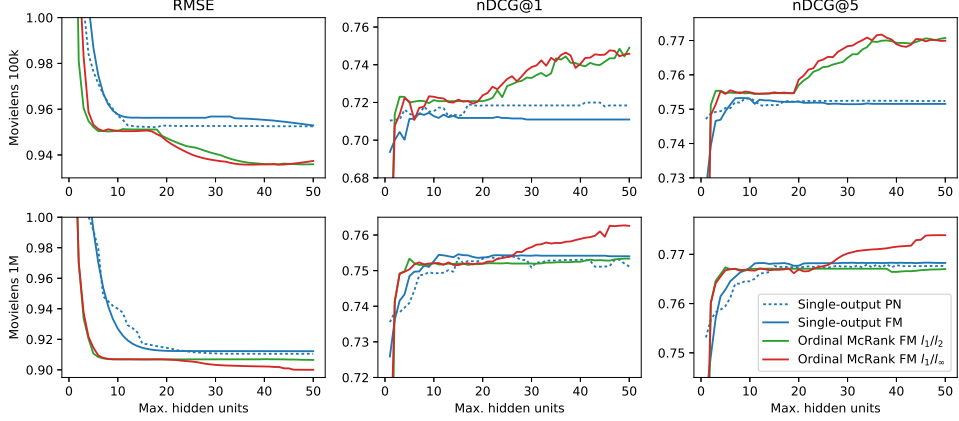

Figure 4: Recommender system experiment: RMSE (lower is better) and nDCG (higher is better).

$i \in [n]$. We form a $n \times m$ matrix $\boldsymbol{Y}$ such that $y_{i,c} = +1$ if $y_i \leq c$ and $-1$ otherwise, and solve

$$\min_{\Omega(\boldsymbol{U}) \leq \tau} \sum_{i=1}^{n} \sum_{c=1}^{m} \ell \left( y_{i,c}, \sum_{\boldsymbol{h} \in \mathcal{H}} \sigma_{\text{ANOVA}}(\boldsymbol{h}, \boldsymbol{x}_i) u_{\boldsymbol{h},c} \right),$$

where $\ell$ is set to the binary logistic loss, in order to be able to produce probabilities. After running Algorithm 1 on that objective for $k$ iterations, we obtain $\boldsymbol{H} \in \mathbb{R}^{k \times d}$ and $\boldsymbol{V} \in \mathbb{R}^{k \times m}$. Because $\boldsymbol{H}$ is shared across all outputs, the only small overhead of using the ordinal McRank reduction, compared to a single-output regression model, therefore comes from learning $\boldsymbol{V} \in \mathbb{R}^{k \times m}$ instead of $\boldsymbol{v} \in \mathbb{R}^k$.

In this experiment, we focus on multi-output factorization machines (FMs), since FMs usually work better than PNs for one-hot encoded data [5]. We show in Figure 4 the RMSE and nDCG (truncated at 1 and 5) achieved when varying $k$ (the maximum number of basis vectors / hidden units).

**Results.** When combined with the ordinal McRank reduction, we found that $l_1/l_2$ and $l_1/l_\infty-$ constrained multi-output FMs substantially outperform single-output FMs and PNs on both RMSE and nDCG measures. For instance, on MovieLens 100k and 1M, $l_1/l_\infty-$constrained multi-output FMs achieve an nDCG@1 of 0.75 and 0.76, respectively, while single-output FMs only achieve 0.71 and 0.75. Similar trends are observed with nDCG@5. We believe that this reduction is more robust to ranking performance measures such as nDCG thanks to its modelling of the expected relevance.

# 7 Conclusion and future directions

We defined the problem of learning multi-output PNs and FMs as that of learning a 3-way tensor whose slices share a common basis. To obtain a convex optimization objective, we reformulated that problem as that of learning an infinite but row-wise sparse matrix. To learn that matrix, we developed a conditional gradient algorithm with corrective refitting, and were able to provide convergence guarantees, despite the non-convexity of the basis vector (hidden unit) selection step.

Although not considered in this paper, our algorithm and its analysis can be modified to make use of stochastic gradients. An open question remains whether a conditional gradient algorithm with provable guarantees can be developed for training deep polynomial networks or factorization machines. Such deep models could potentially represent high-degree polynomials with few basis vectors. However, this would require the introduction of a new functional analysis framework.

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
