[Supplementary Material]

# Supplementary material

## A  Convex multi-class loss functions

Table 3: Examples of convex multi-class loss functions $\ell(y, \boldsymbol{o}) \in \mathbb{R}$, where $y \in [m]$ is the correct label and $\boldsymbol{o} \in \mathbb{R}^m$ is a vector of predicted outputs.

| Loss | $\ell(y, \boldsymbol{o})$ | $\rho_c(y, \boldsymbol{o})$ |
|---|---|---|
| Multi-class logistic | $\log(1 + \sum_{c \neq y} \exp(o_c - o_y))$ | $\frac{\exp(o_c - o_y)}{\sum_{l=1}^m \exp(o_l - o_y)}$ |
| Smoothed multi-class hinge | $\log(1 + \sum_{c \neq y} \exp(1 + o_c - o_y))$ | $\frac{\exp(1[c \neq y] + o_c - o_y)}{\sum_{l=1}^m \exp(1[l \neq y] + o_l - o_y)}$ |
| Multi-class squared hinge | $\sum_{c \neq y} \max(1 + o_c - o_y, 0)^2$ | $2 \max(1 + o_c - o_y, 0)$ |

The gradient w.r.t. $\boldsymbol{o}$, denoted $\nabla \ell(y, \boldsymbol{o}) \in \mathbb{R}^m$, can be computed by

$$\nabla \ell(y, \boldsymbol{o}) = \sum_{c \neq y} \rho_c(y, \boldsymbol{o})(\boldsymbol{e}_c - \boldsymbol{e}_y),$$

where $\boldsymbol{e}_c \in \mathbb{R}^m$ is a vector whose $c^{\text{th}}$ element is 1 and other elements are 0. For the smoothed multi-class hinge loss and the multi-class squared hinge loss, see [27] and [6], respectively.

## B  Proofs

### B.1  Finite support of an optimal solution (Proposition 1)

**General case.** We first state a result that holds for arbitrary activation function $\sigma$ (sigmoid, ReLu, etc...). The main idea is to use the fact that the penalties considered in Table 1 are atomic [8]. Then, we can equivalently optimize (2) over the convex hull of a set of atoms and invoke Carathéodory's theorem for convex hulls.

Let $\phi_{\boldsymbol{h}}(\boldsymbol{X})$ be an $n$-dimensional vector whose $i^{\text{th}}$ element is $\sigma(\boldsymbol{h}^{\text{T}} \boldsymbol{x}_i)$. Let us define the sets

$$\mathcal{A} := \{\boldsymbol{e_h} \boldsymbol{v}^{\text{T}} : \boldsymbol{h} \in \mathcal{H}, \boldsymbol{v} \in \mathcal{V}\} \subset \mathbb{R}^{|\mathcal{H}| \times m} \quad \text{and} \quad \mathcal{B} := \{\phi_{\boldsymbol{h}}(\boldsymbol{X}) \boldsymbol{v}^{\text{T}} : \boldsymbol{h} \in \mathcal{H}, \boldsymbol{v} \in \mathcal{V}\} \subset \mathbb{R}^{n \times m},$$

where we define the set $\mathcal{V}$ as follows:

- $l_1$ case: $\mathcal{V} := \{s \, \boldsymbol{e}_c : s \in \{-1, 1\}, c \in [m]\}$
- $l_1/l_2$ case: $\mathcal{V} := \{\boldsymbol{v} \in \mathbb{R}^m : \|\boldsymbol{v}\|_2 = 1\}$
- $l_1/l_\infty$ case: $\mathcal{V} := \{-1, 1\}^m$.

Then (2) is equivalent to

$$\min_{\boldsymbol{U} \in \mathbb{R}^{|\mathcal{H}| \times m}} \sum_{i=1}^n \ell\left(y_i, \sum_{\boldsymbol{h} \in \mathcal{H}} \phi_{\boldsymbol{h}}(\boldsymbol{X})_i \, \boldsymbol{u_h}\right) \quad \text{s.t.} \quad \Omega(\boldsymbol{U}) \leq \tau$$

$$= \min_{\boldsymbol{U} \in \mathbb{R}^{|\mathcal{H}| \times m}} \sum_{i=1}^n \ell\left(y_i, \sum_{\boldsymbol{h} \in \mathcal{H}} \phi_{\boldsymbol{h}}(\boldsymbol{X})_i \, \boldsymbol{u_h}\right) \quad \text{s.t.} \quad \boldsymbol{U} \in \tau \cdot \text{conv}(\mathcal{A})$$

$$= \min_{\boldsymbol{O} \in \mathbb{R}^{n \times m}} \sum_{i=1}^n \ell(y_i, \boldsymbol{o}_i) \quad \text{s.t.} \quad \boldsymbol{O} \in \tau \cdot \text{conv}(\mathcal{B}),$$

where $\text{conv}(\mathcal{S})$ is the convex hull of the set $\mathcal{S}$. The matrices $\boldsymbol{U}$ and $\boldsymbol{O}$ are related to each other by

$$\boldsymbol{U} = \sum_{\boldsymbol{h} \in \mathcal{H}} \sum_{\boldsymbol{v} \in \mathcal{V}} \theta_{\boldsymbol{h}, \boldsymbol{v}} \boldsymbol{e_h} \boldsymbol{v}^{\text{T}} \quad \text{and} \quad \boldsymbol{O} = \sum_{\boldsymbol{h} \in \mathcal{H}} \sum_{\boldsymbol{v} \in \mathcal{V}} \theta_{\boldsymbol{h}, \boldsymbol{v}} \phi_{\boldsymbol{h}}(\boldsymbol{X}) \boldsymbol{v}^{\text{T}},$$

for some $\boldsymbol{\theta} \in \mathbb{R}^{|\mathcal{H}| \times m}$ such that $\theta_{\boldsymbol{h},\boldsymbol{v}} \geq 0 \; \forall \boldsymbol{h} \in \mathcal{H}, \forall \boldsymbol{v} \in \mathcal{V}$ and $\sum_{\boldsymbol{h} \in \mathcal{H}} \sum_{\boldsymbol{v} \in \mathcal{V}} \theta_{\boldsymbol{h},\boldsymbol{v}} = 1$. By Carathéodory's theorem for convex hulls, there exists $\boldsymbol{\theta}$ with at most $nm + 1$ non-zero elements. Because elements of $\mathcal{A}$ are matrices with a single non-zero row, $\boldsymbol{U}$ contains at most $nm + 1$ non-zero rows (hidden units).

**Case of $l_1$ constraint and squared activation.** When $\sigma(a) = a^2$, given $\boldsymbol{U}$ s.t. $\|\boldsymbol{U}\|_1 \leq \tau$, the $c^{\text{th}}$ output can be written as

$$\sum_{\boldsymbol{h} \in \mathcal{H}} \sigma(\boldsymbol{h}^{\mathrm{T}} \boldsymbol{x}) u_{\boldsymbol{h},c} = \sum_{\boldsymbol{h} \in \mathcal{H}} (\boldsymbol{h}^{\mathrm{T}} \boldsymbol{x})^2 u_{\boldsymbol{h},c} = \boldsymbol{x}^{\mathrm{T}} \left( \sum_{\boldsymbol{h} \in \mathcal{H}} u_{\boldsymbol{h},c} \boldsymbol{h} \boldsymbol{h}^{\mathrm{T}} \right) \boldsymbol{x} =: \boldsymbol{x}^{\mathrm{T}} \boldsymbol{W}_c \boldsymbol{x}.$$

Following [5, Lemma 10], the nuclear norm of a symmetric matrix $\boldsymbol{M} \in \mathbb{R}^{d \times d}$ can be defined by

$$\|\boldsymbol{M}\|_* = \min_{\boldsymbol{\lambda} \in \mathbb{R}^d, \boldsymbol{P} \in \mathbb{R}^{d \times d}} \sum_{j=1}^{d} |\lambda_j| \, \|\boldsymbol{p}_j\|_2^2 \quad \text{s.t.} \quad \boldsymbol{M} = \sum_{j=1}^{d} \lambda_j \boldsymbol{p}_j \boldsymbol{p}_j^{\mathrm{T}}$$

and the minimum is attained by the eigendecomposition $\boldsymbol{M} = \sum_{j=1}^{d} \lambda_j \boldsymbol{p}_j \boldsymbol{p}_j^{\mathrm{T}}$ and $\|\boldsymbol{M}\|_* = \|\boldsymbol{\lambda}\|_1$.

Therefore, we can always compute the eigendecomposition of each $\boldsymbol{W}_c$ and use the eigenvectors as hidden units and the eigenvalues as output layer weights. Moreover, this solution is feasible, since eigenvectors belong to $\mathcal{H}$ and since the $l_1$ norm of all eigenvalues is minimized. Since a matrix can have at most $d$ eigenvalues, we can conclude that $\boldsymbol{U}$ has at most $dm$ elements. Combined with the previous result, $\boldsymbol{U}$ has at most $\min(nm + 1, dm)$ non-zero rows (hidden units).

For the $l_1/l_2$ and $l_1/l_\infty$ penalties, we cannot make this argument, since applying the eigendecomposition might increase the penalty value and therefore make the solution infeasible.

## B.2 Convergence analysis (Theorem 1)

In this section, we include a convergence analysis of the conditional gradient algorithm with multiplicative approximation in the linear minimization oracle. The proof follows mostly from [15] with a trick inspired from [1] to handle multiplicative approximations. Finally, we also include a detailed comparison with the analysis of GECO [26] and Block-FW [18].

We focus on constrained optimization problems of the form

$$\min_{\boldsymbol{x} \in \mathcal{D}} f(\boldsymbol{x}),$$

where $f$ is convex and $\beta$-smooth w.r.t. $\Omega$ and $\mathcal{D} := \{\boldsymbol{x} : \Omega(\boldsymbol{x}) \leq \tau\}$.

**Curvature and smoothness constants.** The convergence analysis depends on the following standard curvature constant

$$C_{f,\mathcal{D}} := \sup_{\substack{\boldsymbol{x},\boldsymbol{s} \in \mathcal{D} \\ \gamma \in [0,1] \\ \boldsymbol{y} = \boldsymbol{x} + \gamma(\boldsymbol{s} - \boldsymbol{x})}} \frac{2}{\gamma^2} \left( f(\boldsymbol{y}) - f(\boldsymbol{x}) - \langle \boldsymbol{y} - \boldsymbol{x}, \nabla f(\boldsymbol{x}) \rangle \right).$$

Intuitively, this is a measure of non-linearity of $f$: the maximum deviation between $f$ and its linear approximations over $\mathcal{D}$. The assumption of bounded $C_{f,\mathcal{D}}$ is closely related to a smoothness assumption on $f$. Following [15, Lemma 7], for any choice of norm $\Omega$, $C_{f,\mathcal{D}}$ can be upper-bounded by the smoothness constant $\beta$ as

$$C_{f,\mathcal{D}} \leq \mathrm{diam}_\Omega(\mathcal{D})^2 \beta.$$

Using $\mathcal{D} = \{\boldsymbol{x} : \Omega(\boldsymbol{x}) \leq \tau\}$, we obtain

$$\mathrm{diam}_\Omega(\mathcal{D}) = \sup_{\boldsymbol{x},\boldsymbol{y} \in \mathcal{D}} \Omega(\boldsymbol{x} - \boldsymbol{y}) \leq \sup_{\boldsymbol{x},\boldsymbol{y} \in \mathcal{D}} \Omega(\boldsymbol{x}) + \Omega(\boldsymbol{y}) \leq 2\tau$$

and therefore

$$C_{f,\mathcal{D}} \leq 4\tau^2 \beta. \tag{7}$$

**Linear duality gap.** Following [15], we define the linear duality gap

$$g_\mathcal{D}(\boldsymbol{x}) := \max_{\boldsymbol{s} \in \mathcal{D}} \langle \boldsymbol{x} - \boldsymbol{s}, \nabla f(\boldsymbol{x}) \rangle.$$

Since $f$ is convex and differentiable, we have that

$$f(s) \geq f(x) + \langle s - x, \nabla f(x) \rangle. \tag{8}$$

Let us define the primal error

$$h_{\mathcal{D}}(x) := f(x) - \min_{x \in \mathcal{D}} f(x).$$

Minimizing (8) w.r.t. $s \in \mathcal{D}$ on both sides we obtain

$$g_{\mathcal{D}}(x) \geq h_{\mathcal{D}}(x).$$

Hence $g_{\mathcal{D}}(x)$ can be used as a certificate of optimality about $x$.

**Bounding progress.** Let $x \in \mathcal{D}$ be the current iterate and $y = x + \gamma(s - x)$ be our update. The definition of $C_{f,\mathcal{D}}$ implies

$$f(y) \leq f(x) + \gamma \langle s - x, \nabla f(x) \rangle + \frac{\gamma^2}{2} C_{f,\mathcal{D}}.$$

We now use that $s$ is obtained by an exact linear minimization oracle (LMO)

$$s = \operatorname*{argmin}_{s \in \mathcal{D}} \langle s, \nabla f(x) \rangle$$

and therefore $\langle s - x, \nabla f(x) \rangle = -g_{\mathcal{D}}(x)$. Combined with $g_{\mathcal{D}}(x) \geq h_{\mathcal{D}}(x)$, we obtain

$$f(y) \leq f(x) - \gamma h_{\mathcal{D}}(x) + \frac{\gamma^2}{2} C_{f,\mathcal{D}}.$$

Subtracting $\min_{x \in \mathcal{D}} f(x)$ on both sides, we finally get

$$h_{\mathcal{D}}(y) \leq (1 - \gamma) h_{\mathcal{D}}(x) + \frac{\gamma^2}{2} C_{f,\mathcal{D}}.$$

**Primal convergence.** Since we use a fully-corrective variant of the conditional gradient method, our algorithm enjoys a convergence rate at least as good as the variant with fixed step size. Following [15, Theorem 1] and using (7), for every $t \geq 1$, the iterates satisfy

$$f(x^{(t)}) - \min_{x \in \mathcal{D}} f(x) \leq \frac{2 C_{f,\mathcal{D}}}{t + 2} \leq \frac{8 \tau^2 \beta}{t + 2}.$$

Thus, we can obtain an $\epsilon$-accurate solution if we run the algorithm for $t \geq \frac{8\tau^2\beta}{\epsilon} - 2$ iterations.

**Linear minimization with multiplicative approximation.** We now extend the analysis to the case of approximate linear minimization. Given $x \in \mathcal{D}$, we assume that an approximate LMO outputs a certain $s \in \mathcal{D}$ such that

$$\langle -s, \nabla f(x) \rangle \geq \nu \max_{s' \in \mathcal{D}} \langle -s', \nabla f(x) \rangle,$$

for some multiplicative factor $\nu \in (0, 1]$ (higher is more accurate). Since $x$ and $y = x + \gamma(s - x)$ are in $\mathcal{D}$, we have like before

$$f(y) \leq f(x) + \gamma \langle s - x, \nabla f(x) \rangle + \frac{\gamma^2}{2} C_{f,\mathcal{D}}.$$

Following the same trick as [1, Appendix B], we now absorb the multiplicative factor $\nu$ in the constraint

$$\langle -s, \nabla f(x) \rangle \geq \max_{s' \in \mathcal{D}'} \langle -s', \nabla f(x) \rangle,$$

where we defined $\mathcal{D}' := \{x : \Omega(x) \leq \tau\nu\} = \nu\mathcal{D}$ (i.e., the ball is shrunk by a factor $\nu$). We therefore obtain $\langle s - x, \nabla f(x) \rangle \leq -g_{\mathcal{D}'}(x)$. Similarly as before, this implies that

$$f(y) \leq f(x) - \gamma h_{\mathcal{D}'}(x) + \frac{\gamma^2}{2} C_{f,\mathcal{D}}.$$

Subtracting $\min_{x \in \mathcal{D}'} f(x)$ on both sides, we get

$$h_{\mathcal{D}'}(y) \leq (1 - \gamma) h_{\mathcal{D}'}(x) + \frac{\gamma^2}{2} C_{f,\mathcal{D}}.$$

We thus get that iterate $\boldsymbol{x}^{(t)}$ satisfies $\boldsymbol{x}^{(t)} \in \mathcal{D}$ and

$$f(\boldsymbol{x}^{(t)}) \leq \min_{\boldsymbol{x} \in \mathcal{D}'} f(\boldsymbol{x}) + \frac{8\tau^2\beta}{t+2}.$$

We can therefore obtain an $\boldsymbol{x}^{(t)} \in \mathcal{D}$ such that $f(\boldsymbol{x}^{(t)}) - \min_{\boldsymbol{x} \in \mathcal{D}'} f(\boldsymbol{x}) \leq \epsilon$ if we run our algorithm for $t \geq \frac{8\tau^2\beta}{\epsilon} - 2$ iterations with constraint parameter $\tau$ and multiplicative factor $\nu$. Put differently, we can obtain an $\boldsymbol{x}^{(t)} \in \frac{1}{\nu}\mathcal{D}$ such that $f(\boldsymbol{x}^{(t)}) - \min_{\boldsymbol{x} \in \mathcal{D}} f(\boldsymbol{x}) \leq \epsilon$ if we run our algorithm for $t \geq \frac{8\tau^2\beta}{\epsilon\nu^2} - 2$ iterations with constraint parameter $\frac{\tau}{\nu}$ and multiplicative factor $\nu$.

**Comparison with the analysis of GECO.** GECO [26] is a greedy algorithm with fully-corrective refitting steps for learning a sparse vector from possibly infinitely-many features, similarly to our algorithm. However, unlike our algorithm, GECO does not constrain the norm of its iterates (i.e., there is no parameter $\tau$), which can lead to severe overfitting in practice. Following [26, Theorem 1], GECO obtains a certain $\boldsymbol{x}^{(t)}$ (unbounded) such that

$$f(\boldsymbol{x}^{(t)}) - f(\boldsymbol{x}) \leq \epsilon \quad \forall \boldsymbol{x}, \forall t \geq \frac{2\|\boldsymbol{x}\|_1^2\beta}{\epsilon\nu^2} - 1. \tag{9}$$

In comparison, for the $l_1$-constrained case, our algorithm learns an $\boldsymbol{x}^{(t)}$ such that $\|\boldsymbol{x}^{(t)}\|_1 \leq \frac{\tau}{\nu}$ and

$$f(\boldsymbol{x}^{(t)}) - \min_{\|\boldsymbol{x}\|_1 \leq \tau} f(\boldsymbol{x}) \leq \epsilon \quad \forall t \geq \frac{8\tau^2\beta}{\epsilon\nu^2} - 2.$$

We see that our algorithm and GECO have similar guarantees, with the difference that GECO does not constrain its iterates.

GECO was used to learn single-output polynomial networks in [20]. Combining (9) together with $\|\boldsymbol{x}\|_\infty\|\boldsymbol{x}\|_0 \geq \|\boldsymbol{x}\|_1$, it was shown that GECO can learn the parameters $\boldsymbol{x}^{(t)}$ (unbounded) of a single-output polynomial network with $l_\infty$ unit ball constraint and squared activation such that

$$f(\boldsymbol{x}^{(t)}) - \min_{\|\boldsymbol{x}\|_\infty \leq 1} f(\boldsymbol{x}) \leq \epsilon \quad \forall \boldsymbol{x}, \forall t \geq \frac{2\|\boldsymbol{x}\|_0^2\beta}{\epsilon\nu^2} - 1.$$

However, if we run our algorithm with an $l_1$ constraint, it can learn an $\boldsymbol{x}^{(t)}$ such that $\|\boldsymbol{x}^{(t)}\|_1 \leq \frac{1}{\nu}$ and

$$f(\boldsymbol{x}^{(t)}) - \min_{\|\boldsymbol{x}\|_\infty \leq 1} f(\boldsymbol{x}) \leq f(\boldsymbol{x}^{(t)}) - \min_{\|\boldsymbol{x}\|_1 \leq 1} f(\boldsymbol{x}) \leq \epsilon \quad \forall t \geq \frac{8\beta}{\epsilon\nu^2} - 2.$$

Clearly, our algorithm with an $l_1$ constraint uses fewer iterations than GECO for learning polynomial networks with $l_\infty$ unit ball constraint and more than $\|\boldsymbol{x}\|_0 = 3$ hidden units.

**Comparison with the analysis of Block-FW.** [18] analyze a block Frank-Wolfe method with "multiplicative" approximations in the linear minimization oracle. However, they require a different condition, namely:

$$\langle \boldsymbol{x} - \boldsymbol{s}, \nabla f(\boldsymbol{x}) \rangle \geq \kappa \cdot \max_{\boldsymbol{s}' \in \mathcal{D}} \langle \boldsymbol{x} - \boldsymbol{s}', \nabla f(\boldsymbol{x}) \rangle$$
$$\Leftrightarrow \langle -\boldsymbol{s}, \nabla f(\boldsymbol{x}) \rangle \geq \kappa \cdot \max_{\boldsymbol{s}' \in \mathcal{D}} \langle -\boldsymbol{s}', \nabla f(\boldsymbol{x}) \rangle + \langle \boldsymbol{x}, \nabla f(\boldsymbol{x}) \rangle(\kappa - 1),$$

for some $\kappa \in (0, 1]$. Under this condition, they show that the algorithm converges to an $\epsilon$-approximate solution in $\mathcal{O}(\frac{1}{\epsilon})$ iterations. A disadvantage of the above condition is that it contains an additive term that depends on the current iterate $\boldsymbol{x}$ and so it is difficult to give guarantees on $\kappa$ in general.

# C   Computing an optimal solution of the linearized subproblem ($l_1/l_\infty$ case)

We describe how to compute an optimal hidden unit $\boldsymbol{h}^\star$ in the $l_1/l_\infty$ case, albeit with exponential complexity in $m$. Because of its exponential complexity in $m$ (the number of outputs), clearly, this method should only be used to evaluate other (polynomial-time) algorithms.

Recall that we want to solve

$$\max_{\boldsymbol{h} \in \mathcal{H}} f_1(\boldsymbol{h}) = \sum_{c=1}^{m} |\boldsymbol{h}^{\mathrm{T}}\boldsymbol{\Gamma}_c\boldsymbol{h}|.$$

Now, if we knew the sign $s_c := \mathrm{sign}(\boldsymbol{h}^{\star \mathrm{T}} \boldsymbol{\Gamma}_c \boldsymbol{h}^\star)$, we could rewrite the problem as

$$\max_{\boldsymbol{h} \in \mathcal{H}} f_1(\boldsymbol{h}) = \sum_{c=1}^{m} s_c \boldsymbol{h}^{\mathrm{T}} \boldsymbol{\Gamma}_c \boldsymbol{h} = \boldsymbol{h}^{\mathrm{T}} \left( \sum_{c=1}^{m} s_c \boldsymbol{\Gamma}_c \right) \boldsymbol{h},$$

whose optimal solution is the dominant eigenvector of the symmetric matrix $\sum_{c=1}^{m} s_c \boldsymbol{\Gamma}_c$. The idea is then simply to find the dominant eigenvector for all possible $2^m$ sign vectors and choose the eigenvector that achieves largest objective value.

## D  Implementation details

In practice, penalized formulations are more convenient to handle than constrained ones. Here, we discuss why we can safely replace constrained formulations by penalized formulations in the refitting step. We use the output layer refitting objective as an example. It is well known that there exists $\lambda > 0$ such that this objective is equivalent to

$$\min_{\boldsymbol{V} \in \mathbb{R}^{t \times m}} F(\boldsymbol{V}, \boldsymbol{H}^{(t)}) + \lambda \Omega(\boldsymbol{V}).$$

Unfortunately, the relation between $\tau$ and $\lambda$ is a priori unknown. However, it is easy to see that the constant factor $\tau$ in (3) is absorbed by the output layer in our refitting step. This means that we need to know the actual value of $\tau$ for the refitting step but not for the hidden unit selection step. As long as we compute a full regularization path, we may therefore use a penalized formulation in a practical implementation. We do so for both refitting objectives we discussed.

For both refitting objectives, we use FISTA, an accelerated projected gradient method with $\mathcal{O}(1/t^2)$ convergence rate, where $t$ is the iteration number. We set the maximum number of iterations to 1000 and the stopping criterion's tolerance to $10^{-3}$.

## E  Datasets

For our multi-class experiments, we used the following four publicly available datasets [12].

| Name | $n$ | $d$ | $m$ |
|---|---|---|---|
| segment | 2,310 | 19 | 7 |
| vowel | 528 | 10 | 11 |
| satimage | 4,435 | 36 | 6 |
| letter | 15,000 | 16 | 26 |

For recommender system experiments, we used the following two publicly available datasets [14].

| Name | $n$ | $d$ | $m$ |
|---|---|---|---|
| Movielens 100k | 100,000 (ratings) | 2,625 = 943 (users) + 1,682 (movies) | 5 |
| Movielens 1M | 1,000,209 (ratings) | 9,940 = 6,040 (users) + 3,900 (movies) | 5 |

The task is to predict ratings between 1 and 5 given by users to movies, i.e., $y \in \{1, \ldots, 5\}$. The design matrix $\boldsymbol{X}$ was constructed following [25]. Namely, for each rating $y_i$, the corresponding $\boldsymbol{x}_i$ is set to the concatenation of the one-hot encodings of the user and item indices. Hence the number of samples $n$ is the number of ratings and the number of features is equal to the sum of the number of users and items. Each sample contains exactly two non-zero features. It is known that factorization machines are equivalent to matrix factorization when using this representation [25].

# F Additional experimental results

## F.1 Multi-class squared hinge loss results

We also compared the multi-class logistic (ML) loss to the multi-class squared hinge (MSH) loss. The MSH loss achieves comparable test accuracy to the ML loss. However, it can often be much faster to train, since it does not require expensive exponential and logarithm calculations.

Table 4: Comparison between multi-class squared hinge (MSH) and logistic (ML) losses.

| Constraint | Loss | Conditional gradient (full refitting) | | | | Conditional gradient (output-layer refitting) | | | |
|---|---|---|---|---|---|---|---|---|---|
| | | segment | vowel | satimage | letter | segment | vowel | satimage | letter |
| $l_1$ (#units) | MSH | 96.01 (21) | 87.83 (8) | 89.98 (22) | 92.03 (130) | 95.67 (21) | 79.13 (25) | 88.99 (21) | 91.25 (149) |
| | ML | 96.71 (41) | 87.83 (12) | 89.80 (25) | 92.29 (150) | 97.05 (20) | 80.00 (21) | 89.71 (40) | 91.01 (139) |
| $l_1/l_2$ (#units) | MSH | 96.01 (15) | 86.96 (8) | 90.25 (12) | 91.57 (94) | 95.67 (25) | 85.22 (19) | 89.98 (50) | 92.03 (149) |
| | ML | 96.71 (40) | 89.57 (15) | 89.08 (18) | 91.81 (106) | 96.36 (21) | 85.22 (15) | 89.71 (50) | 92.24 (150) |
| $l_1/l_\infty$ (#units) | MSH | 95.84 (16) | 85.22 (18) | 89.80 (29) | 92.27 (149) | 97.05 (28) | 86.09 (33) | 88.90 (24) | 91.20 (119) |
| | ML | 96.71 (24) | 86.96 (15) | 88.99 (20) | 92.35 (149) | 96.19 (16) | 86.96 (41) | 89.35 (41) | 91.68 (128) |

## F.2 Full vs. output layer refitting comparison

In this experiment, we compare output layer refitting with full refitting of both the hidden and output layers. Empirically, we observe that full refitting does not always outperform output layer refitting in terms of objective value but it does so in terms of test accuracy.

Figure 5: Relative objective difference from best (top) and multi-class test set accuracy values (bottom) when performing output layer refitting (dashed) and full, non-convex refitting (solid), optimizing a penalized $l_1/l_2$ objective with $\lambda = 0.1$.