[Reviews · NeurIPS 2017]

Reviewer 1



This paper proposes an extension of factorization machines and polynomial networks for enabling them to predict vector valued functions. This is an important extension of an expressive and powerful class of models. The learning algorithm provided is sound and the experiments show it works well enough. The results clearly show a significant performance improvement over the single output version of these models, which, in my opinion, is enough to show the usefulness of the proposed approach.

Reviewer 2



The authors extend factorization machines and polynomial networks to the multi-output setting casting their formulation as a 3-way tensor decomposition. Experiments on classification and recommendation are presented. I enjoyed reading the paper and although the model itself seems a somewhat incremental extension with respect to [5], the algorithmic approach is interesting and the theoretical results are welcome. I would like to see runtimes of the proposed model compared to baselines, kernelized in particular, to highlight the benefits of the proposed method. Besides, the classification experiments could use at least one large dataset, the largest used is letter (N=15,000), again it will serve to highlight the benefits of the proposed model. I understand that a polynomial SVM with quadratic kernel is a natural baseline for the proposed model but I am curious about the results with more popular kernels choices, i.e, 3rd degree polynomial or RBF. The recommender system experiment needs baselines to put the results from single-output and ordinal McRank FMs in context. Minor comments: - line 26, classical approaches such as?

Reviewer 3



The paper proposes a model to learn a multi-output function of two sets of features such as the ones occurring in recommender systems. The idea of the model is to reduce its number of parameters by using a last layer where each unit (corresponding to each output) shares a basis of internal representations. The idea seems sound and well explained.